# Assessment of inter-city transport of particulate matter in the Beijing-Tianjin-Hebei region

Xing Chang[1], Shuxiao Wang[1,2], Bin Zhao[3], Siyi Cai[1], and Jiming Hao[1,2]

[1] State Key Joint Laboratory of Environment Simulation and Pollution Control, School of Environment, Tsinghua University, Beijing 100084, China

[2] State Environmental Protection Key Laboratory of Sources and Control of Air Pollution Complex, Beijing 100084, China

[3] Joint Institute for Regional Earth System Science and Engineering and Department of Atmospheric and Oceanic Sciences, University of California, Los Angeles, CA 90095, USA

*Correspondence to*: Shuxiao Wang [shxwang@tsinghua.edu.cn]

and Bin Zhao [zhaob1206@ucla.edu]

## Abstract

The regional transport of $PM_{2.5}$ plays an important role in the air pollution of the Beijing-Tianjin-Hebei (BTH) region in China. However, previous studies on regional transport of $PM_{2.5}$ mainly aim at province level, which is insufficient for the development of an optimal joint $PM_{2.5}$ control strategy. In this study, we calculate $PM_{2.5}$ inflows and outflows through the administrative boundaries of three major cities in the BTH region, i.e. Beijing, Tianjin and Shijiazhuang, using the WRF (Weather Research and Forecasting model) -CMAQ (Community Multiscale Air Quality) modelling system. The monthly average inflow fluxes indicate the major directions of $PM_{2.5}$ transport. For Beijing, the $PM_{2.5}$ inflow fluxes from Zhangjiakou (on the northwest) and Baoding (on the southwest) constitute 57% of the total in winter, and Langfang (on the southeast) and Baoding constitute 73% in summer. Based on the net $PM_{2.5}$ fluxes and their vertical distributions, we find there are three major transport pathways in the BTH region: the Northwest-Southeast pathway in winter (at all levels below 1000 m), the Southeast-Northwest pathway in summer (at all levels below 1000 m), and the Southwest-Northeast pathway both in winter and in summer (mainly at 300 – 1000 m). In winter, even if surface wind speeds are low, the transport at above 300 m could still be strong. Among the three pathways, the Southwest-Northeast happens along with $PM_{2.5}$ concentrations 30% and 55% higher than the monthly average in winter and summer, respectively. Analysis of two heavy pollution episodes in January and July in Beijing show a much stronger (8-16 times) transport than the monthly average, emphasizing the joint air pollution control of the cities located on the transport pathways, especially during heavy pollution episodes.

Key words: $PM_{2.5}$ flux; inter-city transport; CMAQ model; Beijing-Tianjin-Hebei region

## 1. Introduction

The Beijing-Tianjin-Hebei (BTH) region, one of the most developed regions in China, is suffering from severe pollution of particulate matter with diameter less than 2.5 μm ($PM_{2.5}$). According to the monitoring data from China National Environmental Monitoring Centre (http://www.cnemc.cn/), the average $PM_{2.5}$ concentrations of the BTH region in 2013, 2014 and 2015 were 106 μg/m³, 93 μg/m³ and 77 μg/m³, respectively, which far exceeded the 35 μg/m³ standard in China. The high $PM_{2.5}$ concentrations have adverse impacts on visibility (Zhao et al., 2011b) as well as human health (Zhang et al., 2013), and thus may cause a large economic loss

(Mu and Zhang, 2013). Therefore, it is urgent to reduce the $PM_{2.5}$ concentration in the BTH region.
Emissions from one city can substantially affect the $PM_{2.5}$ pollution in another city under particular
meteorology conditions by the transport process. For example, some studies showed that emissions from
outside Beijing can contribute to 28-70% of the ambient $PM_{2.5}$ concentration in Beijing (An et al., 2007; Streets
et al., 2007; BJEPB, 2015; Wang et al., 2014b). A number of approaches have been applied to evaluate the
inter-city transport of $PM_{2.5}$ and its effect on local air quality. The backward trajectory, such as the HYSPLIT
model (Stein et al., 2015), is one of the most commonly used methods. This method can provide the most
probable transport trajectory of the air mass before it arrives at a target location; however, it cannot quantify
the inter-city transport of $PM_{2.5}$. Another commonly used method is the sensitivity analysis based on Euler 3-
D models, such as CMAQ (Community Multiscale Air Quality model), which is done by calculating the
change in concentration due to a change in emissions. This method includes the Brute Force Method (e.g.
Wang et al., 2015), the decoupled direct method (DDM, (Itahashi et al., 2012)), and the Response Surface
Model (RSM, (Zhao et al., 2015)). These methods are all based on a chemical transport model, so that the
physical and chemical processes can both be well considered. However, the sensitivity of $PM_{2.5}$ concentration
in the target city to emissions from the source city is not necessarily the same as the contribution of transport
process, because of the non-linear relationships between emissions and concentrations(Kwok et al., 2015).
Based on the simulated meteorology field and air pollutant concentrations, the inter-city transport of $PM_{2.5}$ can
be simply expressed by the $PM_{2.5}$ flux through city boundaries. Compared to the preceding methods, the flux
approach can give direct and quantitative assessment of the transport of pollutants without a heavy calculation
burden. This approach has been widely applied to assess the large scale transport of air pollutants, such as
inter-continent transports (Berge and Jakobsen, 1998; In et al., 2007). There are also studies that evaluated the
pollutant transport on a regional scale (Jenner and Abiodun, 2013; Wang et al., 2009); some of which focused
on the BTH region (An et al., 2012; Wang et al., 2010). In those studies, the boundaries for flux calculations
are at the province level. However, in China, the air pollution control strategy is formulated and implemented
at the city level. Moreover, most previous studies regarding $PM_{2.5}$ transports in the BTH region focused on
Beijing. In recent years, however, under the policy of "integrating development of BTH region", the air quality
in Tianjin and the cities in Hebei Province are being increasingly emphasized. Therefore, a systematic
assessment of the $PM_{2.5}$ flux at the city level in the BTH region is needed.
In this study, we select Beijing, Tianjin and Shijiazhuang as target cities, and calculate the inter-city $PM_{2.5}$
transport fluxes through the administrative boundaries between the target cities and the neighboring prefecture-
level cities, based on the WRF (Weather Research and Forecasting model) –CMAQ modeling system. The
PM$_{2.5}$ transport pathway in the BTH region are identified based on the PM$_{2.5}$ transport flux results.

## 2.  Methodology

### 2.1.  Emission inventory

A multiscale emission inventory is used in this study. For the region outside China mainland, we use the MIX
emission inventory (Li et al., 2017) of the year 2010. For the China mainland other than the BTH region, we
adopt a gridded emission inventory of 2012 developed in our previous study (Cai et al., 2017). For the BTH
region, we develop a bottom-up emission inventory of 2012. A unit-based approach is used for power plants,
iron and steel plants, and cement plants (Zhao et al., 2008). Emission factor approach is used for other sectors
(Fu et al., 2013; Zhao et al., 2013b). In particular, emissions in Beijing are updated from the bottom-up
inventory developed by Tsinghua University and Beijing Municipal Research Institute of Environmental
Protection (BJEPB, 2010; Zhao et al., 2011a). The emissions of major pollutants in each city are shown in
Table 1. Methods for the biogenic emissions, the VOC speciation and the spatial and temporal allocation of
emissions are consistent with our previous study (Zhao et al., 2013a). The spatial distributions of emissions
are shown in Fig. S1 (see the Supplementary Information (SI)).

### 2.2.  WRF-CMAQ model configuration

We establish a one-way, triple nesting domain in the WRF-CMAQ model to simulate the meteorology and air
pollutant fields, as shown in Fig. 1. Domain 1 covers mainland China and part of East Asia and Southeast Asia
at a grid resolution of 36 km × 36 km; Domain 2 covers the eastern China at a grid resolution of 12 km × 12
km; Domain 3 covers the BTH region at a grid resolution of 4 km × 4 km, which is the target area of this study.
The simulation periods are January and July 2012 representing the winter and summer time, respectively.
For the WRF (version 3.7) model, 23 sigma levels are selected for the vertical grid structure. The top layer
pressure is 100 mb at approximately 15 km. The National Center for Environmental Prediction (NCEP)'s Final
Operational Global Analysis data with a horizontal resolution of 1°×1° at every 6 h are used to generate the
first guess field. The NCEP's Automated Data Processing (ADP) data are used in the objective analysis scheme.
The major physics options are the Kain-Fritsch cumulus scheme, the Pleim-Xiu land surface model, the ACM2
planetary boundary layer (PBL) scheme, the Morrison double-moment cloud microphysics scheme, and the
Rapid Radiative Transfer Model (RRTM) longwave and shortwave radiation scheme. The Meteorology-
Chemistry Interface Processor (MCIP) version 3.3 is applied to convert the WRF output data to a format
required by CMAQ.
We use CMAQv5.0.2 to simulate the air quality field. The CMAQ model is configured with the AERO6
aerosol module and the CB-05 gas-phase chemical mechanism. The default profile is used to generate the
boundary condition of the first domain, and the simulation results of the outer domains provide the boundary
conditions for the inner domains. The simulation begins five days ahead of each month to minimize the impact
of initial condition.
The model predicted meteorology and $PM_{2.5}$ concentrations are compared with observation data. The results
are shown in the Supplementary Information (SI). The simulations agree well with observations. Most of the
indices are within the benchmarks suggested by Emery et al. (2001). We evaluate simulated $PM_{2.5}$
concentrations against observations at 5 sites located in Domain 3, i.e., Beijing, Shijiazhuang, Xianghe,
Xinglong and Yucheng (see Fig. 1), as shown in Table 2. The time series of simulated and observed $PM_{2.5}$
concentrations are shown in Fig. 2. It can be seen that the variation trends of $PM_{2.5}$ are well reproduced both
in January and in July for all 5 sites. The average $PM_{2.5}$ concentrations are slightly underestimated in January,
while the underestimation is larger in July in most sites, especially in Beijing and Xinglong. However, the
MFB and MFE indices in January and July for the domain all fall inside the "criteria" benchmark value
suggested by Boylan and Russell (2006). To understand the reason of the underestimation, it is necessary to
evaluate the simulation results of major components of $PM_{2.5}$. Given that we have no observations of $PM_{2.5}$
components in 2012 in the BTH region, we additionally simulate the air quality in July and August in 2013,
and compare it with the $PM_{2.5}$ component observations at several sites (see details in the SI). Generally, the
underestimation of total $PM_{2.5}$ in the summer time mainly comes from the underestimation of organic carbon
(OC) and sulfate. The default CMAQ tends to underestimate secondary organic aerosol to a large extent,
especially in summer when photochemical reactions are active, which is a common problem of most widely
used chemical transport models (Simon and Bhave, 2012; Heald et al., 2005; Zhao et al., 2016). The lack of
aqueous oxidation of $SO_2$ by $NO_2$ (Wang et al., 2016), and $SO_2$ oxidation at dust surface (Fu et al., 2016) may
partly account for the underestimation of sulfate. The underestimation of sulfate also partly explains the
overestimation of nitrate. Moreover, the biases of $PM_{2.5}$ major components in the current study fall in a similar
range with other studies in the BTH region (Wang et al., 2015; Wang et al., 2014a; Wang et al., 2011; Zhao et
al., 2017; Liu et al., 2016) (See details in the SI). In conclusion, the biases of simulated meteorological field
and $PM_{2.5}$ concentrations fall in a reasonable range. The modelling results can be used for further studies.

### 2.3. $PM_{2.5}$ flux calculation

The $PM_{2.5}$ flux in this study stands for the mass of $PM_{2.5}$ that flow through a particular vertical surface in a
particular period of time. The vertical surface extends from the ground to a particular vertical level along the
boundary of two regions (Fig. 3(a)). However, the model can only provide three-dimensional discrete wind
field and $PM_{2.5}$ concentration field. Therefore, the vertical surface through which the flux is calculated is
discretized to several vertical grid cells, as is illustrated in Fig. 3(b) and detailed in the next paragraph. In this
case, the expression of $PM_{2.5}$ flux can be written as
$$Flux = \sum_{i=1}^{h} \sum_{l} LH_i c\vec{v} \cdot \vec{n} \tag{1}$$

where $l$ is the boundary line of two regions; $h$ is the top layer; $L$ is the grid width; $H_i$ is the height between
layer $i$ and $i-1$; $c$ is the concentration of $PM_{2.5}$ at the vertical grid cell; $\vec{v}$ is the wind vector, and $\vec{n}$ is the
normal vector of the vertical grid cell. The variables in the expression can be obtained from the output of the
models. We choose the 9[th] layer above the ground (about 1000 m) as the top layer, because most of the $PM_{2.5}$
transport between regions happens inside the boundary layer (Shi et al., 2008). Even though the transport could
happen above the boundary layer, the influence of such transport on the near ground concentrations is less
important because the vertical mixing above the boundary layer is weaker.
Beijing and Tianjin are two most important and developed megacities in the BTH region. Shijiazhuang is the
capital city, and also one of the most developed and polluted cities in Hebei province. Therefore, we choose
these three cities as the target cities for flux calculation. In order to accurately distinguish the transport from
different adjacent cities and to understand the net $PM_{2.5}$ inflow of a city as a whole, all the administrative
boundaries between the target city and the adjacent cities are chosen as the boundary line. The boundary lines
are separated to different segments by neighbor cities, and the fluxes are calculated separately for each segment.
The locations of the three target cities and their neighbors are shown in Fig. 1. Note that there is a small area
surrounded by Beijing and Tianjin that belongs to the city of Langfang, so the boundaries between Beijing and
Tianjin, Beijing and Langfang, and Tianjin and Langfang are each separated into two segments. To distinguish
them, we add the relative location of the boundary to the neighbor city's name, like "Beijing (N)" and "Beijing
(S)".
The flux varies every now and then, depending on the wind direction. The polluted air mass may flow in, affect
the local air quality and flow out subsequently in a short time, so that the flux may offset each other during the
integration. Therefore, to characterize the intensity of interactions between two regions as well as the general
impact of $PM_{2.5}$ transport, three indices are chosen in regard to the flux calculation, that is the inflow flux,
outflow flux and net flux.

## 3.  Results and discussion
### 3.1.  Characteristics of the inter-city $PM_{2.5}$ transport in January
The monthly inflow, outflow and net fluxes through each boundary segment of the three target cities are shown
in Fig. 4, from which we can get an overview of the transport in a relatively long period of time. We treat the
fluxes as positive if $PM_{2.5}$ flows into the target cities, and vice versa. Therefore, the positive total net fluxes in
Beijing and Shijiazhuang reveal that the $PM_{2.5}$ inflows of these two cities generally exceed the outflows, and
that these cities act as a "sink" of $PM_{2.5}$. This is possibly due to the unique terrain of Beijing and Shijiazhuang.
These two cities are both half-surrounded by western and northern mountains, while major emissions of $PM_{2.5}$
lie to the south and east. Consequently, pollutants are easily trapped in the bulging part of the plain if there is
a weak wind from the south or the east. The trapped pollutants are either scavenged by wet deposition without
flowing out, or diluted by strong vertical convection due to the strong northwestern wind brought by the cold
front and thus flow out of the boundary layer. In contrast, Tianjin behaves as a "source" of $PM_{2.5}$ flux.
Furthermore, a probe into the detailed inflow, outflow and net fluxes through each boundary segment of the
three cities may help us understand the extent to which the cities interact with their neighbors. For Beijing, in
winter, the inflow fluxes mainly come from Zhangjiakou (on the northwest) and Baoding (on the southwest),
and the outflows go to Chengde (on the northeast) and Langfang (on the southeast) more than the others. For
Tianjin, Langfang (on the northwest) and Tangshan (on the northeast) contribute most of the inflow fluxes,
and the Bohai sea (on the southeast) and Tangshan again receive the major outflow fluxes. Shijiazhuang acts
differently from Beijing and Tianjin. The inflow and outflow fluxes through all the four boundary segments
are considerably strong, where Xingtai (on the south) and Baoding (on the northeast) contributing relatively
more to inflow and outflow fluxes, respectively.
$PM_{2.5}$ fluxes may vary with height. We calculate the vertical distribution of net flux through each boundary
segment to see at what level the transport mainly occurs. The results are shown in Fig. 5 (a), (c) and (e). The
fluxes of each vertical layer in the CMAQ model are shown separately, and the approximate elevation of each
layer is marked on the left. Generally, the total flowing intensity is stronger at higher levels for all three cities,
while the major contributor varies with layers. If we add up the net fluxes through all boundary segments
(shown by the narrow bars with an envelope line), we can see that the "sink" behavior of Beijing is mainly
contributed by the total net fluxes at 400 to 600 m where contribution from Baoding (on the southwest) exhibits
a rapid increase with height. Similarly, the total net flux for Tianjin shows a peak value near 600 m where
Tangshan (on the northeast) receives much more outflow than it does near the ground. Total net flux for
Shijiazhuang shows a peak value near 400 m where Hengshui (on the east) and Xingtai (on the south) have
dominant contributions.
In order to better understand the general image of the transport characteristics in the BTH region, we display
the net flux results on a map, using arrows to represent the net flux direction and intensity. The result of January
is shown in Fig. 6(a). Bigger arrow represents larger flux, and white and black arrows denote fluxes at the
lower (layer 1-5 in the model, from the ground to about 310 m) and upper (layer 6-9 in the model, from about
310 m to about 1000 m) layers, respectively. From the map we can identify two key $PM_{2.5}$ transport pathways
in the BTH region in January: the Northwest-Southeast pathway (Zhangjiakou -> Beijing -> Langfang ->
Tianjin -> The Bohai Sea) and the Southwest-Northeast pathway (Xingtai -> Shijiazhuang -> Baoding ->
Beijing -> Chengde). The former is related to the prevailing wind direction brought by winter monsoon in the
BTH region, and happens at both lower layers and higher layers. The latter happens mainly at higher layers.
According to the Ekman Spiral, wind speed is much higher at the upper level of the boundary layer (Holton
and Hakim, 2012), so that pollutants can travel a longer distance during their lifetime. Assuming that the
emission height of each city is similar, we believe that a $PM_{2.5}$ inflow at higher altitude origins more probably
from a farther source. From this point of view, the $PM_{2.5}$ flow of the Southwest-Northeast pathway at higher
levels may consist of a relatively long range transport. In winter time, the southwest wind field usually occurs
after the passage of a cold high pressure, when the wind speed is low and the sky is clear. Such air condition
traps less upward infrared radiation at night, which helps to enhance the air stability, or even causes

temperature inversion. Moreover, the southwest wind also brings moisture, leading to the formation of fog, which may enhance the aqueous reaction to form more particles. Therefore, southwest wind is usually accompanied by pollution. The Southwest-Northeast transport pathway should be intensely considered during the winter time in the BTH region. In contrast, the northwest wind usually comes during the passage of a cold high pressure, with a relatively high wind speed both at lower and higher levels bringing dry, cold and clean air from the non-polluted area. The large fluxes from northwest are more likely due to the strong winds rather than the high $PM_{2.5}$ levels.

## 3.2. Characteristics of the inter-city $PM_{2.5}$ transport in July

We conduct the same calculation in July to probe into the transport characteristics in summer. The monthly average inflow, outflow and net fluxes are shown in Fig. 4. Similar to January, total net fluxes are positive (more inflow than outflow) for Beijing and Shijiazhuang, and negative (more outflow than inflow) for Tianjin, though the magnitude is much higher than that in January. In detail, the inflow fluxes for Beijing mainly come from Langfang (on the southeast) and Baoding (on the southwest), and the outflow fluxes mainly go to Chengde (on the northeast) and Zhangjiakou (on the northwest). For Tianjin, Bohai sea (on the east) and Tangshan (on the northeast) contribute a large part of the inflow, and Langfang (on the northwest) and Tangshan receive most of the outflow fluxes. The transport directions for Beijing and Tianjin in July are quite different from those in January. However, for Shijiazhuang, all of the four directions (Shanxi, Baoding, Hengshui and Xingtai) still contribute comparable amount of inflow and outflow fluxes, where inflows from Xingtai (on the south) and Hengshui (on the east) are slightly larger.

Fig. 5 (b), (d) and (f) display the vertical distributions of monthly average net fluxes with respect to the three cities in July. For Beijing, the total net fluxes are positive at all levels, which are different from those in January. The major contributor, Baoding and Langfang, show different behaviors. Net flux from Baoding is nearly zero near the ground, but increases rapidly with height, while the net flux from Langfang (including both Langfang (N) and Langfang (S)) is significant at all levels, and is largest at medium height. These phenomena are tied to the wind speed and direction at different heights in the BTH region in summer. The dominant wind direction near the ground is from the southeast. Within the boundary layer, the wind will rotate clockwise and become stronger at higher levels according to Ekman Spiral (Holton and Hakim, 2012). Langfang and Baoding are located to the southeast and southwest of Beijing, respectively. The increase of wind speed and the rotation of

wind direction will constantly enhance the $PM_{2.5}$ transport from southwest, but could contribute oppositely to the transport from southeast, causing a local maximum in middle layers. For Tianjin, the overall outflow happens mainly at levels below 600 m, where the outflow flux mainly goes to Langfang. The inflow flux is dominated by the Bohai Sea at all heights, indicating a cross-sea transport from Shandong or other areas. The vertical distribution of net fluxes for Shijiazhuang is quite similar to that in January, except that Shanxi no more contribute a considerable amount of inflow flux.

We also show the general transport characteristics in the BTH region with arrows on the map, as is shown in Fig. 6(b). Compared with that in winter, the transport at lower layers becomes stronger. We can also figure out two major transport pathways in BTH in July: the Southwest-Northeast pathway (Xingtai -> Shijiazhuang -> Baoding -> Beijing -> Chengde), and the Southeast-Northwest pathway (Bohai -> Tianjin -> Langfang -> Beijing -> Zhangjiakou, and Hengshui -> Shijiazhuang). The latter pathway, which is caused by the summer monsoon, is significant at both lower and upper layers. The pathway from southwest to northeast is only obvious at upper layers. Considering that in summer the vertical mixing is stronger, although the Southwest-Northeast pathway is only active at higher levels, the transport may still affect the near-ground concentration remarkably.

If we put together the transport characteristics in winter and summer, we can see that, aside from the opposite transport pathways brought by the monsoon in different seasons, there is a steady transport pathway from southwest to northeast in the BTH region regardless of the season. This pathway has also been found in some other studies. Wu et al. (2017) analyzed the regional persistent haze events in the BTH region during 1980-2013, and found that southwestern wind field at 925 hPa (~800 m) is a typical meteorology condition. Backward trajectory studies by Zhao et al. (2017) also found a southerly transport pathway during pollution periods in the BTH region. Therefore, the Southwest-Northeast pathway is indeed important in the BTH region. To better understand how the wind and concentration affect the transport fluxes, we calculate the frequency of wind directions and the corresponding wind speed and $PM_{2.5}$ concentration, and plot them as "wind rose" plots. We show the plots of Beijing in Fig. 7 as an example. The plots for the other two cities can be found in SI.

In January, the dominant wind directions near the ground ranges from northwest to northeast. The NNE wind has the highest frequency, while the NW wind has the highest wind speed (Fig. 7(a)). The dominant northern winds reflect the winter monsoon. Although the concentration coming with the northern winds are relatively low because of the low emission rate on that direction(Fig. 7(b)), the high frequency and wind speed also cause

an overall strong transport from the northwest to the southeast. Wind directions and the corresponding concentrations are quite different at the upper layers (Fig. 7(c), (d)). The prevalent northern wind remains (though the dominant directions shift slightly from NNE to NW), and the frequency of southwestern winds is much higher than that at lower layers. Moreover, the $PM_{2.5}$ concentrations that come with southwestern winds are much higher than the other directions. The strong emission in southern Hebei (which lies on the southwest direction of Beijing), especially the elevated source may be responsible for the high concentration from the southwest. Therefore, in January, the dominant northwestern winds account for the Northwest-Southeast pathway at both lower layers and upper layers, while the large emissions on the southwest direction mainly caused the Southwest-Northeast pathway at upper layers.

In July, the dominant wind directions at the lower layer are the southeastern directions, reflecting the summer monsoon (Fig 7(e)), and coincidentally the highest concentrations also come along with the southeastern winds (Fig 7(f)). Emissions from Tianjin, Langfang, and Tangshan may influence Beijing by the southeastern winds. The emission and the wind direction both contribute to the Southeast-Northwest pathway at the lower layers. The high frequency wind directions shift clockwise to the southern directions at the upper layers in July, as is shown in Fig 7(g), and the southwest wind and the southeast wind are both important. Moreover, the directions with high concentrations also shift to both the southwest and the southeast directions (Fig 7(h)). Therefore, in July, the dominant southeastern winds and the emissions on the southeast directions caused the Southeast-Northwest pathway at both the upper and the lower layers. The Southwest-Northeast pathway is a combination result from the southern winds and the emissions, which is different from that in January.

The monthly transport characteristics could bring us inspiration on how the joint control of different cities should be applied. The transport pathway at lower layers suggests that we should primarily control nearby low-level emission sources, while the pathway at upper layers calls for the control over a larger region to the upstream direction.

### 3.3. The daily characteristics of $PM_{2.5}$ transport in Beijing

In addition to the monthly characteristics of $PM_{2.5}$ transports discussed in Section 3.2, we analyzed the daily characteristics in this sector, taking Beijing as an example. Firstly, since different $PM_{2.5}$ concentration may be caused by different meteorology condition, and may also result in different transport flux characteristics in

different days, we first calculate the PM$_{2.5}$ flux during different pollution levels (Fig. 8). We sort the daily data into 6 groups in January and 5 groups in July. The separating points are chosen to be near the 30, 55, 75, 85 and 95 percentiles in January, and the 30, 60, 80, 90 percentiles in July. The groups are denser at higher concentrations to better reveal the details around heavy pollution periods.

In January, the transport becomes stronger when the concentration is higher, but the transport flux decreases in turn when the concentration is the highest. The inflow from Baoding and outflow to Chengde, which are the indicator of the Southwest-Northeast pathway, also rise gradually, followed by a sudden decrease. In July, the situation is similar, though the decrease is less significant. Such result is consistent with Tang et al. (2015) and Zhu et al. (2016) that the Southwest-Northeast transport pathway is more significant when the pollution is still rising.

To reveal the daily characteristics comprehensively, we present the net PM$_{2.5}$ fluxes of Beijing during two heavy-pollution episodes in January and July of 2012 as examples. In January, we choose 17[th] - 19[th], which are the most polluted days in January (the simulated PM$_{2.5}$ daily average concentrations all exceed 200 µg/m$^3$ ). In July, we also choose the period with the highest concentration, i.e. 18[th] to 20[th]. The results are shown in Fig. 9.

The magnitude of net fluxes in January 17[th] and 18[th] (-590 t/day and 688 t/day) is much higher than the monthly average value (139 t/day). For 17[th] Jan, there are some weak outflows mainly to Langfang at lower levels, while stronger inflows from Baoding and Zhangjiakou occur at 300-600m. On 18[th] Jan, fluxes at lower level remain relatively small though the inflow and outflow directions reverse. However, strong inputs from Baoding and Langfang at above 300 m become significantly strong. It can be seen that although the fluxes near the ground are small, the inflow transport can be quite strong at levels above 300 m. Coincidentally, the elevation of the mountains in the northwest of Beijing are commonly higher than 300 m, making it harder for the inflowing PM$_{2.5}$ to flow out. The large amount of PM$_{2.5}$ inflows can only be efficiently blown out to the northeast direction (Chengde, Langfang (N) and Tianjin (N)). These results are consistent with Jiang et al. (2015), who also found a strong southerly input at a high level during a haze episode in winter.

However, in 19[th] January when the concentration reaches the peak, the inflow transport becomes weaker than the previous days, especially for the southwest inflow. The PM$_{2.5}$ experienced a significant inflow from the southwest followed by an accumulation period with little inflow. Therefore, the Southwest-Northeast pathway is of great importance during the first days of this heavy pollution period.

For the day with the highest concentration in July (July 20[th]), the vertical distribution does not show much difference from the average of July (Fig. 5 (f)), except for the magnitude. The fluxes are about 1/5 of the monthly average, or less than 1/10 of that in the heavy-pollution period in January. This result suggests that the heavy pollution in Beijing in 20[th] July is not dominated by the inter-city transport during that very day. However, situations are totally different on 18[th] and 19[th] July (Fig. 7 (d,e)), the days when the simulated $PM_{2.5}$ concentration reaches a high level but is still rising in this pollution episode. The magnitude of fluxes is about 6 times larger than the monthly average, or some 30 times larger than that on 20[th] July. More importantly, the outflow flux is much smaller than the inflow flux contributed mainly by Baoding and Langfang, which correspond to the Southwest-Northeast and Southeast-Northwest pathways respectively. Therefore, we can draw an image about how the $PM_{2.5}$ transport affects the air quality in Beijing during this pollution episode. On 18[th] July, the $PM_{2.5}$ start to flow into Beijing through the Southeast-Northwest and Southwest-Northeast pathways with a very strong flux, but very few of them flow out, causing the accumulative increase of $PM_{2.5}$ concentration. On 20[th] July, the wind field become stable and the transport weakened, but the $PM_{2.5}$ that have flowed in before accumulate to form the heavy pollution. This result indicates that both the Southeast-Northwest and the Southwest-Northeast pathway are important for Beijing during this polluted period, and the emission from outside Beijing should be controlled at least 2 days in advance to reduce the peak concentration. From the discussions above, we can see that $PM_{2.5}$ transport plays an important role in the heavy-pollution periods in Beijing. We further analyze the $PM_{2.5}$ flux data of the three cities day by day, and try to identify the presence of transport pathway for each day in Beijing, based on whether the inflow flux from a certain direction is significantly larger than the others. Finally, 8 days in January and 4 days in July are subject to the transport of Southwest-Northeast pathway, 22 days in January are subject to the transport of Northwest-Southeast pathway, and 8 days in July are subject to the transport of the Southeast-Northwest pathway. In July, there are other 8 days that are subject to both the Southeast-Northwest pathway and the Southwest-Northeast pathway ("SE-NW + SW-NE" for short). Moreover, some days do not show a clear transport direction, which are referred to as "unclassifiable days". We calculate the average simulated concentration for each transport pathway. The results are shown in Table 3.

The days with Southwest-Northeast pathway show the highest $PM_{2.5}$ average concentrations among all days in both January and July. Therefore, the Southwest-Northeast pathway should be the focus of control strategies. In contrast, the Northwest-Southeast pathway tends to happen along with the lowest concentrations in both

seasons. Note that in January, the day with the highest concentration (January 19[th]) is coincidentally identified as the Northwest-Southeast pathway. That day is on the eve of the rapid clearing by the northwest wind (Fig. 2(a)). While the cold front is passing, the heavy polluted air mass is forced to move from northwest to southeast, which cause a significant transport. However, since the pollution brought by such transport usually happen with a strong cold front, the $PM_{2.5}$ concentration will soon become very low (Jia et al., 2008). If we exclude January 19[th] from the Northwest-Southeast pathway days, the average concentration will be only 48.5 μg/m³. In July, the Southeast-Northwest pathway and the Southwest-Northeast pathway happen simultaneously in 8 days. The average concentration is 47.4 μg/m³, the second highest in July, which further emphasizes the importance of the transport from the southwest. In summary, the Southwest-Northeast pathway should be taken great consideration both in January and July, followed by the Southeast-Northwest pathway in July.

Besides the daily variability of $PM_{2.5}$ transport, we also analyzed the diurnal variability brought by the "mountain – plain wind cycle" in summer times in Beijing (Tang et al., 2016). However, because the average plain wind is much stronger than the mountainous wind which is only obvious below 200 m, the fluxes brought by the mountainous wind is much weaker than that by plain wind (Fig. S6). The diurnal variation of winds does not have a significant influence on the direction of the transport fluxes.

## 4. Conclusions

By calculating $PM_{2.5}$ inflow and outflow fluxes through the boundaries between each two prefecture-level cities, this study has shown the major $PM_{2.5}$ input and output directions in winter and summer for Beijing, Tianjin, and Shijiazhuang. For Beijing, the inflow fluxes mainly come from northwest and southwest in winter, and southeast and southwest in summer. For Tianjin, the inflow fluxes are mostly from northwest and northeast in winter, and east and northeast in summer. In Shijiazhuang, however, the four neighboring regions contribute comparable amount of inflow fluxes both in winter and summer.

By analyzing the net $PM_{2.5}$ fluxes and their vertical distribution, we identify several major transport pathways and the height they occur: the Northwest-Southeast pathway in winter (at all levels below 1000 m, but stronger at levels above 300 m), the Southeast-Northwest pathway in summer (at all levels below 1000 m), and the Southwest-Northeast pathway both in winter and in summer (at levels between 300 m and 1000 m). Although the third pathway does not happen as frequently as the other two in corresponding seasons, it is accompanied

by quite high PM$_{2.5}$ concentrations in both seasons. Additionally, the relatively large transport height of this pathway suggests the importance of the long-range transport of PM$_{2.5}$ on air quality. Specially, in winter, even if the wind speed near the ground is low, which we often refer to as "steady" conditions, the transport above 300 m, which is primarily associated with long-range transport, could still be strong. These findings suggest that the joint control for cities on the Southwest-Northeast pathway should be emphasized both in winter and summer.

By analyzing daily transport fluxes in Beijing, we also find that the flux during the days with higher PM$_{2.5}$ concentration is generally higher, but the flux during the top 10% polluted days is smaller. The flux during heavy-pollution episodes is stronger than the monthly average for the two polluted periods investigated in this study. In the heavy pollution episode in summer, PM$_{2.5}$ flows into Beijing and accumulates for two days, leading to a heavy pollution. Therefore, mitigating emissions from a larger area may be essential for the control of ambient PM$_{2.5}$ in Beijing. Moreover, it appears important to control the upstream sources several days ahead to mitigate the PM$_{2.5}$ accumulations, rather than only taking actions when the pollution is already heavy. However, we must note that the two episodes we studied may not represent the general characteristics of heavy-pollution episodes, which requires a more systematic analysis in the future.

The current study has several limitations. First, we only quantify the transport of PM$_{2.5}$ at the boundary of the city, which is not the only way by which transport process may influence the PM$_{2.5}$ concentration in the target city. Other processes include the inter-city transport of gaseous precursors that remain in gaseous phase at the boundary but may convert to secondary PM$_{2.5}$ in the target city. Secondly, the PM$_{2.5}$ transported through the outer boundary is a mixture of different sources that does not only from the neighbor city itself. Although we have obtained a general transport feature in the BTH region which can facilitate a qualitative understanding of where the fluxes are mainly from, the flux approach cannot quantitatively evaluate the contribution from each city in the upstream areas. If we want to overcome these disadvantages, a life-time tracing during the emission, transportation, reaction and deposition processes of PM$_{2.5}$ and its gaseous precursors is needed. Therefore, future studies may combine the flux calculation with the tagging models to overcome these defects. Despite these limitations, the flux approach has indeed proved to be a powerful tool to visually assess the inter-city transport of pollutants.

## Acknowledgments

We hereby express our gratitude to Yangjun Wang from Shanghai University, Jia Xing from Tsinghua University and Jiandong Wang from Max Planck Institute who helped us set up the modelling system and gave us useful suggestions.

This research has been supported by National Science Foundation of China (21625701 & 21521064). The simulations were completed on the "Explorer 100" cluster system of Tsinghua National Laboratory for Information Science and Technology.

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

**Table 1 Summary of the emissions of major pollutants in Beijing, Tianjin and 11 prefecture-level cities**
**in Hebei in 2012**

| Emissions (kt/year) | NOx | SO$_2$ | PM$_{2.5}$ | PM$_{10}$ | BC | OC | NMVOCs | NH$_3$[a] |
|---|---|---|---|---|---|---|---|---|
| **Beijing** | 202 | 120 | 75 | 177 | 9 | 9 | 381 | 52 |
| **Tianjin** | 392 | 287 | 113 | 151 | 17 | 26 | 287 | 45 |
| **Hebei** | 1620 | 1079 | 875 | 1172 | 141 | 221 | 1346 | 628 |
| Shijiazhuang | 270 | 198 | 149 | 203 | 23 | 33 | 230 | 87 |
| Chengde | 84 | 45 | 37 | 49 | 6 | 10 | 56 | 34 |
| Zhangjiakou | 112 | 52 | 41 | 54 | 7 | 11 | 56 | 35 |
| Qinhuangdao | 71 | 39 | 30 | 40 | 5 | 8 | 51 | 22 |
| Tangshan | 266 | 145 | 100 | 135 | 15 | 24 | 181 | 68 |
| Langfang | 79 | 71 | 63 | 86 | 10 | 14 | 100 | 35 |
| Baoding | 158 | 123 | 118 | 155 | 20 | 33 | 202 | 89 |
| Cangzhou | 149 | 121 | 109 | 148 | 17 | 25 | 164 | 67 |
| Hengshui | 79 | 66 | 62 | 84 | 10 | 15 | 92 | 50 |
| Xingtai | 140 | 105 | 77 | 102 | 13 | 21 | 113 | 60 |
| Handan | 213 | 115 | 89 | 117 | 15 | 26 | 148 | 82 |


**Table 2 Comparison of the simulated and observed PM₂.₅ concentrations at five sites.**

| Indices | | Mean OBS | Mean SIM[a] | NMB | NME | MFB | MFE |
|---|---|---|---|---|---|---|---|
| Unit | | μg·m⁻³ | μg·m⁻³ | % | % | % | % |
| | Beijing | 86.0 | 65.2 | -24.2 | 32.2 | | |
| | Shijiazhuang | 193.9 | 170.8 | -11.9 | 45.3 | | |
| **January, 2012** | Xianghe | 132.3 | 85.6 | -35.3 | 44.5 | -19.6 | 19.6 |
| | Xinglong | 39.4 | 38.6 | -2.0 | 42.7 | | |
| | Yucheng | 140.9 | 124.1 | -11.9 | 31.2 | | |
| | Beijing | 68.2 | 35.6 | -47.8 | 49.5 | | |
| | Shijiazhuang | 70.3 | 79.8 | +13.6 | 37.6 | | |
| **July, 2012** | Xianghe | 61.3 | 47.2 | -23.0 | 35.6 | -35.1 | 40.2 |
| | Xinglong | 48.9 | 24.6 | -49.6 | 53.8 | | |
| | Yucheng | 77.3 | 55.2 | -28.6 | 39.6 | | |
| "Criteria" benchmark[b] | | - | - | - | - | ≤±60 | ≤75 |
| "Goal" benchmark[b] | | - | - | - | - | ≤±30 | ≤50 |

a. Average of the days only when observations are available.
b. Benchmarks are suggested by Boylan and Russell (2006).

**Table 3 The mean and maximum simulated PM$_{2.5}$ concentrations in Beijing for all days in January and**
**July and for the days that belong to particular transport pathways.**

| Month | Pathway type | Days | Mean PM$_{2.5}$ conc in Beijing, μg/m$^3$ | Max PM$_{2.5}$ conc in Beijing, μg/m$^3$ |
|---|---|---|---|---|
| Jan | All days | 31 | 65.2 | 270.7 |
| | Southwest-Northeast | 8 | 85.1 | 211.5 |
| | Northwest-Southeast | 22 | 58.6 | 270.7 |
| | Unclassifiable day(s) | 1 | 53.0 | 53.0 |
| Jul | All days | 31 | 35.0 | 94.4 |
| | Southwest-Northeast | 4 | 54.2 | 94.4 |
| | Northwest-Southeast | 5 | 15.4 | 30.9 |
| | Southeast-Northwest | 8 | 29.4 | 53.2 |
| | SW-NE + SE-NW | 8 | 47.4 | 71.0 |
| | Unclassifiable day(s) | 9 | 29.3 | 79.7 |


**Figure Captions**

**Figure 1. The simulation domains used in this study (left) and the map of the Beijing-Tianjin-Heibei region (right). The highlighted cities are the target cities for flux calculation. The red circles show the sites with $PM_{2.5}$ observations. The two sites with green circles have observations of $PM_{2.5}$ chemical components in 2013.**

**Figure 2. Time series of the simulated and observed PM2.5 concentrations in (a) Beijing, (b) Shijiazhuang, (c) Xianghe, (d) Xinglong, and (e) Yucheng.**

**Figure 3. An example of the vertical surface for flux calculation (a) before discretization, and (b) after discretization.**

**Figure 4. The inflow, outflow and net fluxes in January and July for (a) Beijing, (b) Tianjin, and (c) Shijiazhuang.**

**Figure 5. Vertical distribution of net fluxes in January (left) and July (right) for (a-b) Beijing, (c-d) Tianjin, and (e-f) Shijiazhuang**

**Figure 6. The transport fluxes through each boundary segment of the three target cities in January (a) and July (b). The size of the arrows represents the amount of the fluxes, while white and black arrows denote fluxes at the lower (layer 1-5 in the model, from the ground to about 310 m) and upper (layer 6-9 in the model, from about 310 m to about 1000 m) layers, respectively.**

**Figure 7. The wind rose plots showing the frequency of wind speed (a, c, e, g) and $PM_{2.5}$ concentration (b, d, f, h) at different wind directions for Beijing. The ground level and the 7th level (about 450-600 m) in the model are chosen as the representation of lower levels and upper levels. The percentages denote the frequency.**

**Figure 8. $PM_{2.5}$ average flux in different pollution degrees in (a) January and (b) July.**

**Figure 9. $PM_{2.5}$ fluxes during heavy-pollution days in Beijing in January and July: (a) January 17th, (b) January 18th, (c) January 19th, (d) July 18th, (e) July 19th and (f) July 20th..**

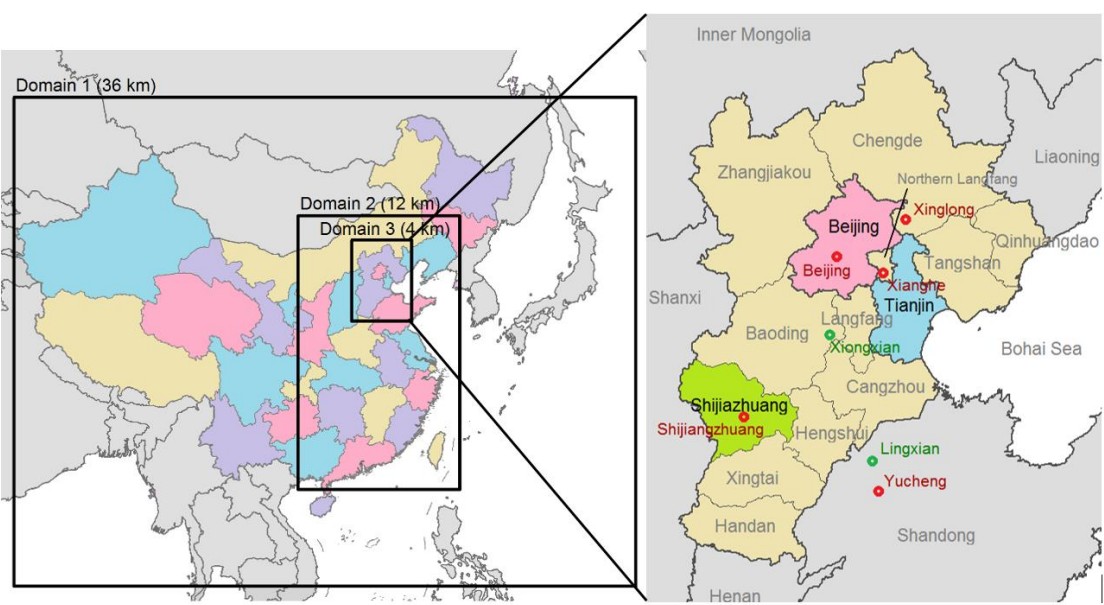


**Figure 1 The simulation domains used in this study (left) and the map of the Beijing-Tianjin-Heibei**
**region (right). The highlighted cities are the target cities for flux calculation. The red circles show the**
**sites with PM$_{2.5}$ observations. The two sites with green circles have observations of PM$_{2.5}$ chemical**
**components in 2013.**

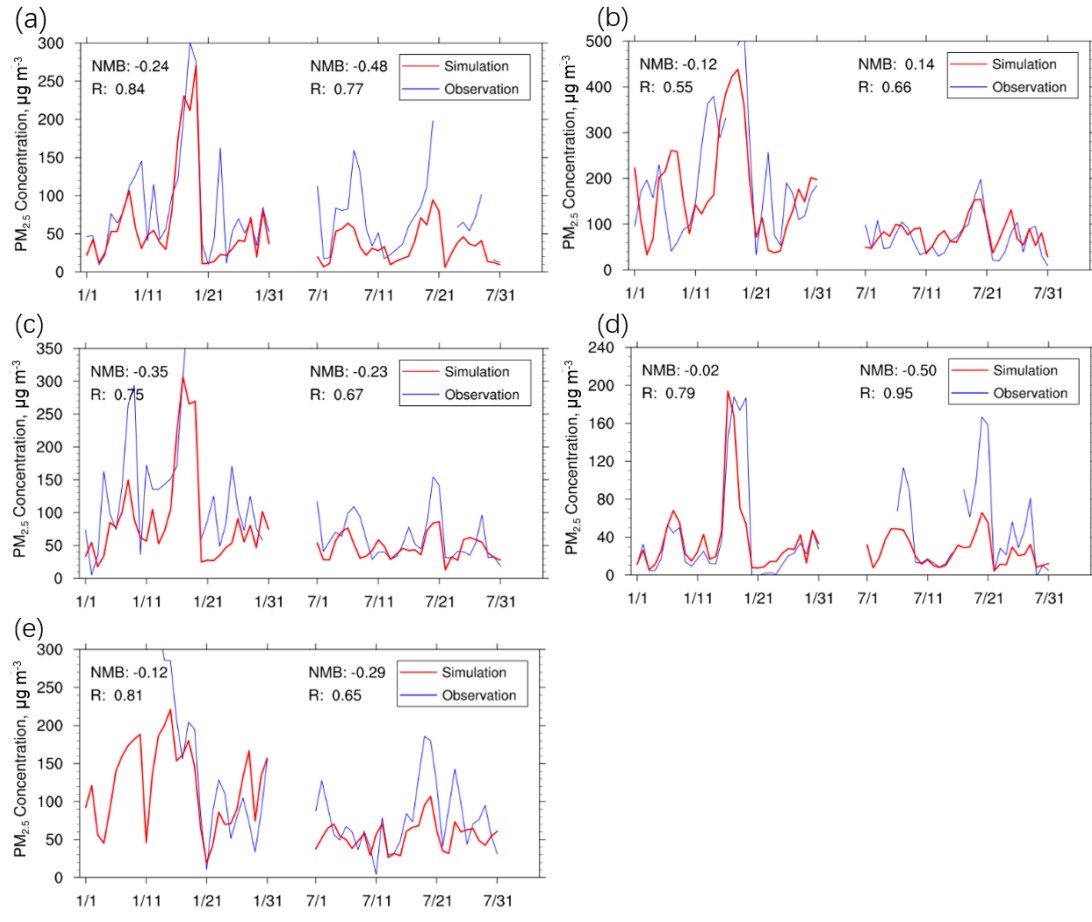


**Figure 2 Time series of the simulated and observed PM2.5 concentrations in (a) Beijing, (b) Shijiazhuang,**

**(c) Xianghe, (d) Xinglong, and (e) Yucheng**


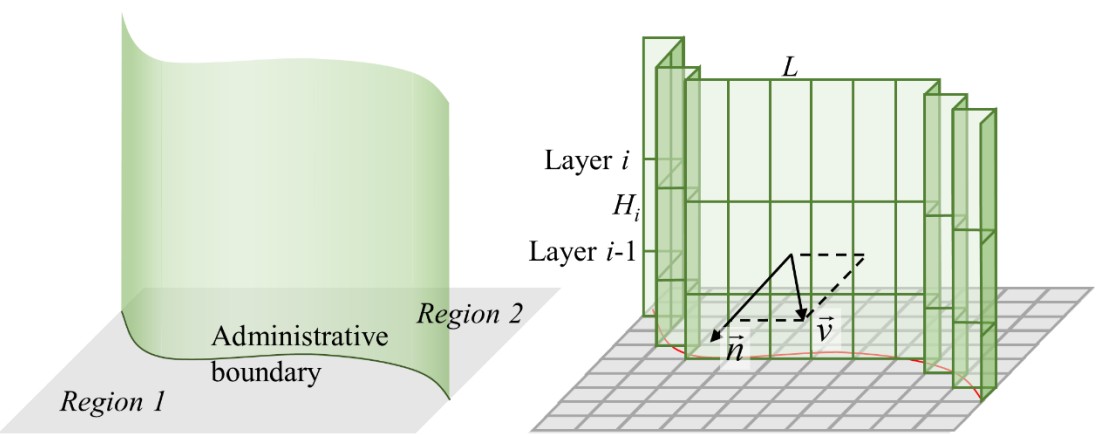


**Figure 3 An example of the vertical surface for flux calculation (a) before discretization, and (b) after**

**discretization.**


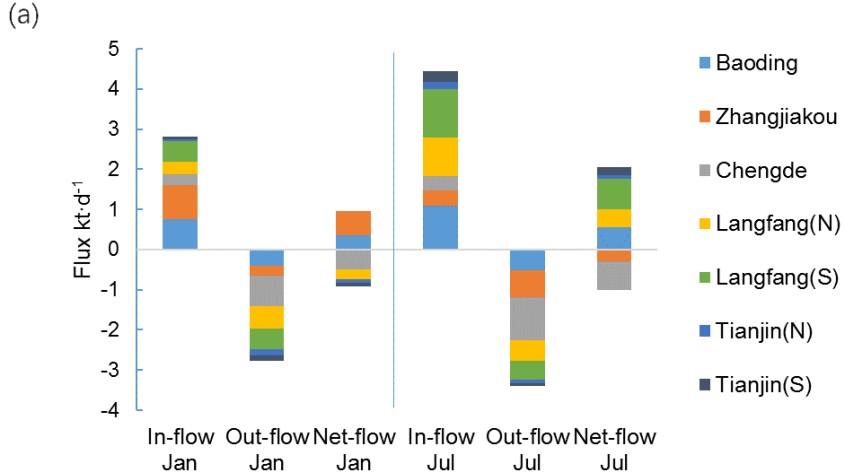

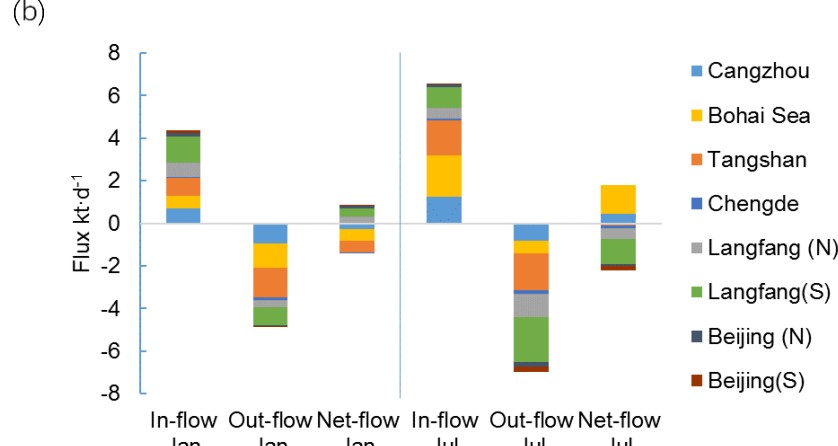

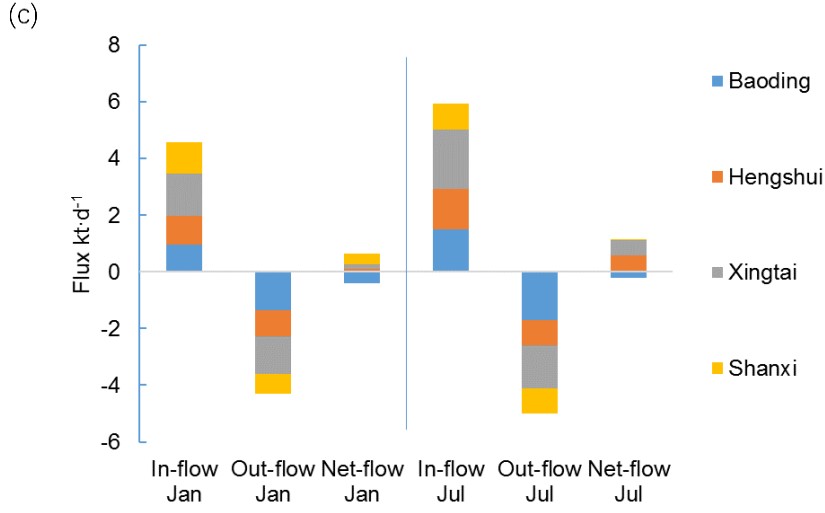


**Figure 4 The inflow, outflow and net fluxes in January and July for (a) Beijing, (b) Tianjin, and (c)**
**Shijiazhuang**

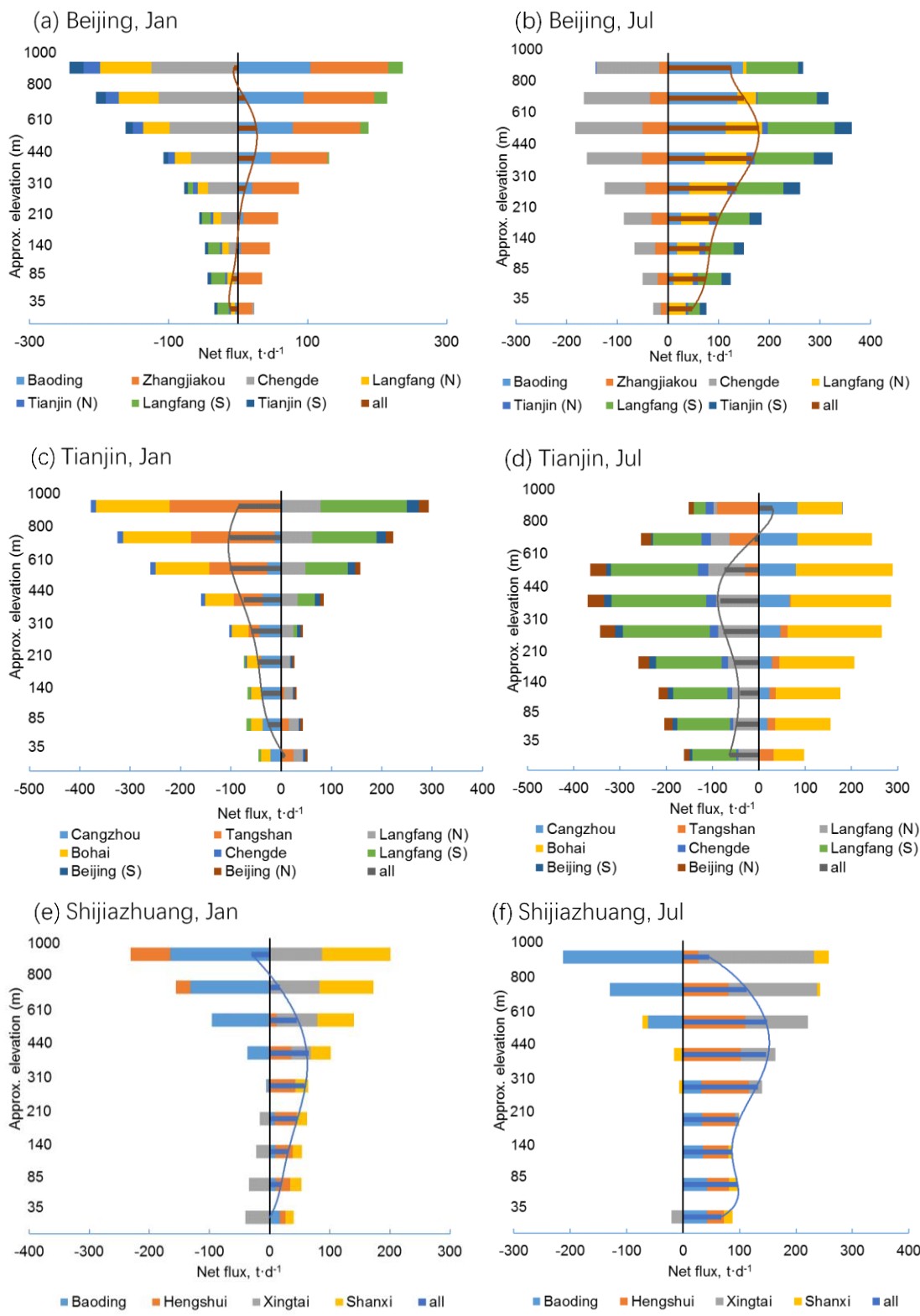

**Figure 5 Vertical distribution of net fluxes in January (left) and July (right) for (a-b) Beijing, (c-d) Tianjin, and (e-f) Shijiazhuang**

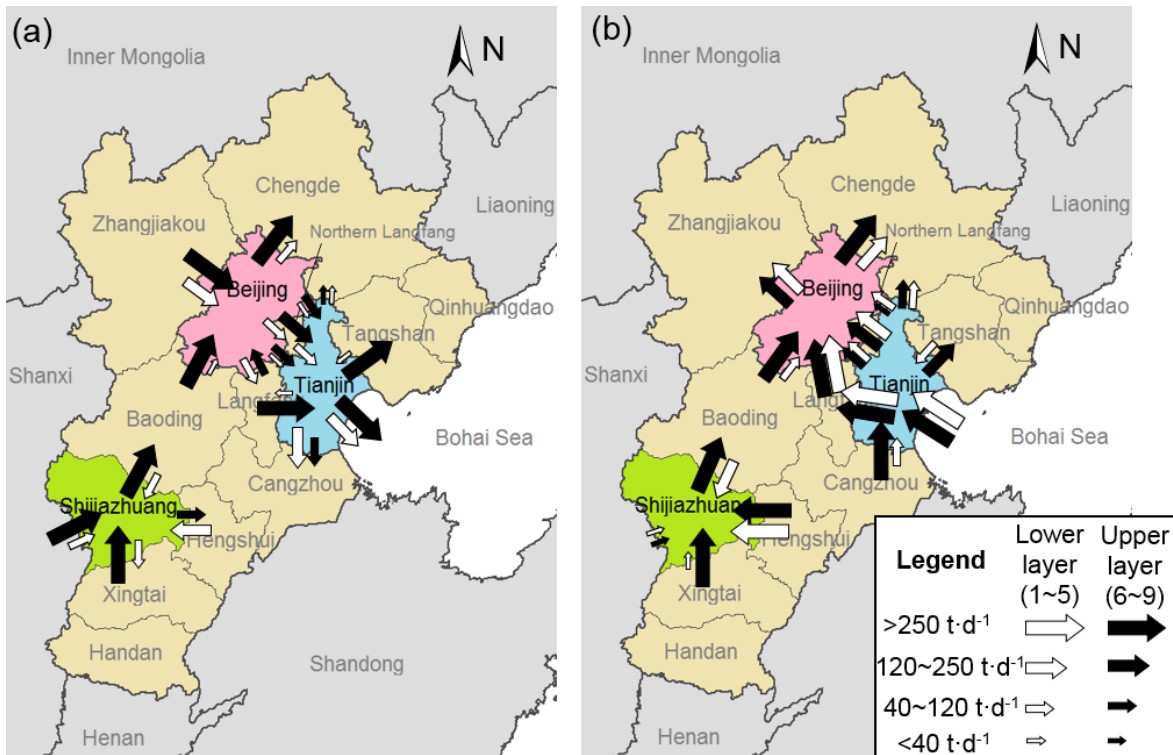

**Figure 6 The transport fluxes through each boundary segment of the three target cities in January (a) and July (b). The size of the arrows represents the amount of the fluxes, while white and black arrows denote fluxes at the lower (layer 1-5 in the model, from the ground to about 310 m) and upper (layer 6-9 in the model, from about 310 m to about 1000 m) layers, respectively.**

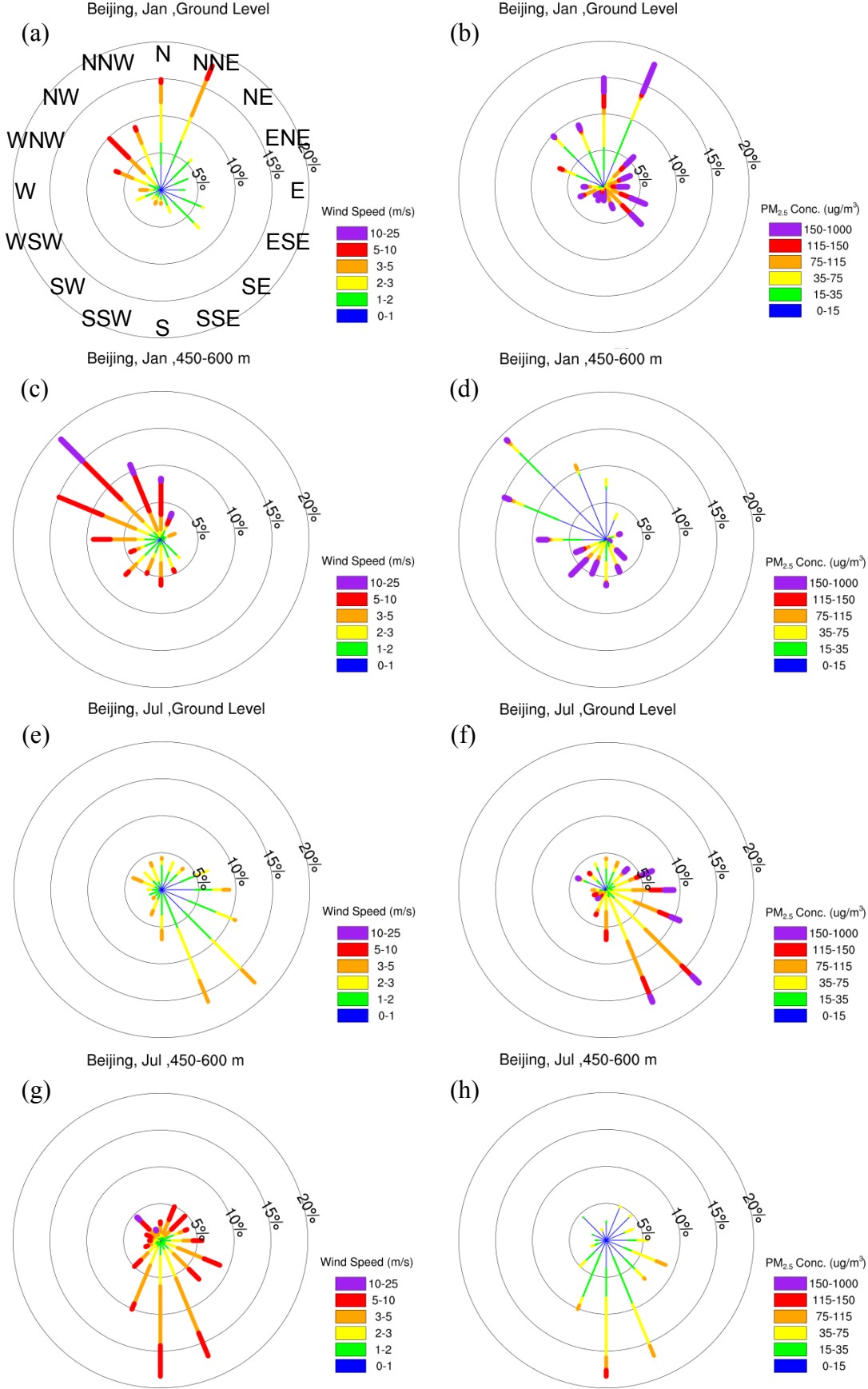


**Figure 7 The wind rose plots showing the frequency of wind speed (a, c, e, g) and PM₂.₅ concentration**

**(b, d, f, h) at different wind directions for Beijing. The ground level and the 7[th] level (about 450-600 m)**
**in the model are chosen as the representation of lower levels and upper levels. The percentages denote**
**the frequency.**

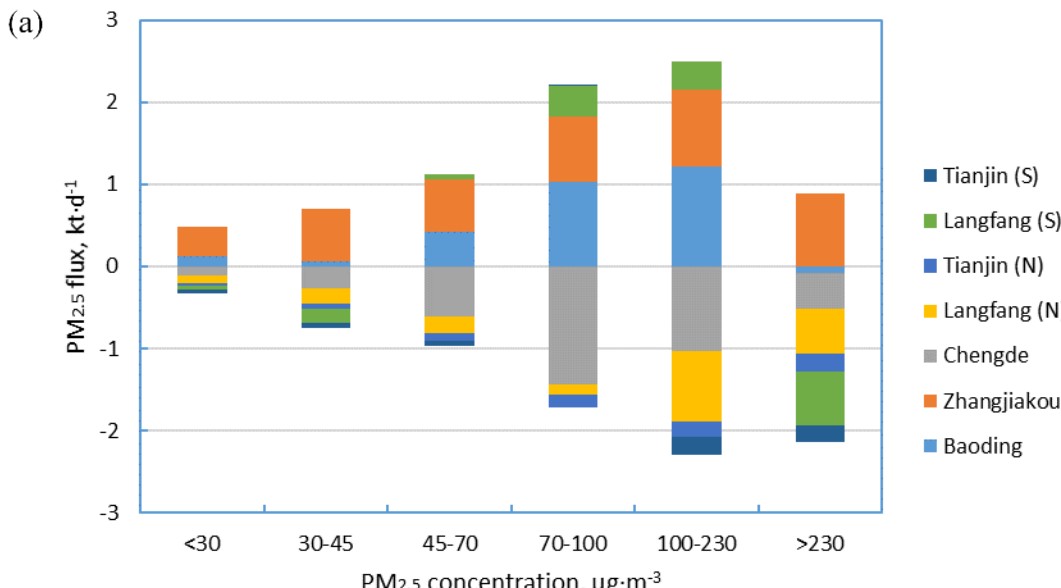

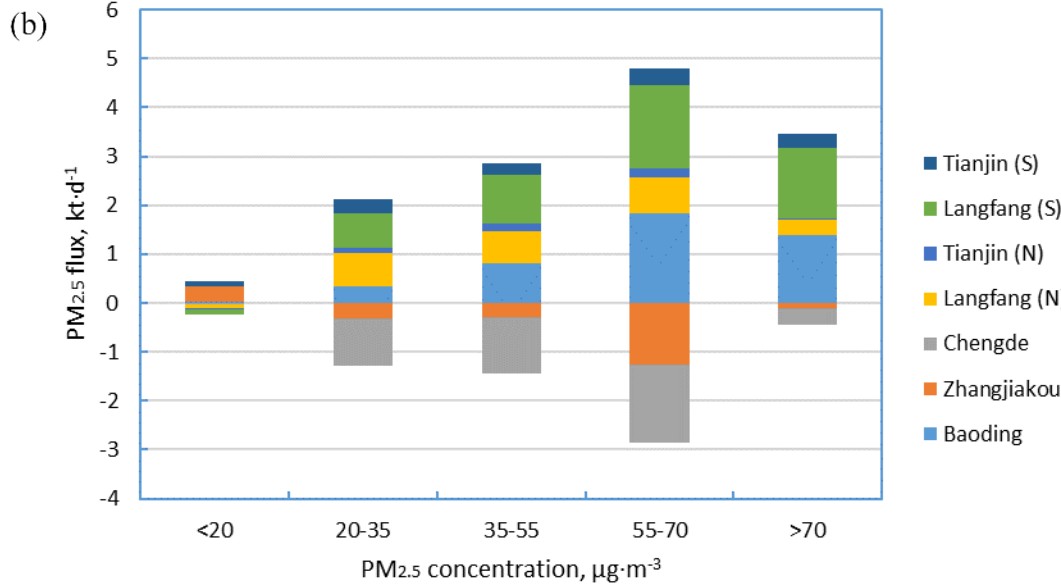


**Figure 8 PM$_{2.5}$ average flux in different pollution degrees in (a) January and (b) July.**

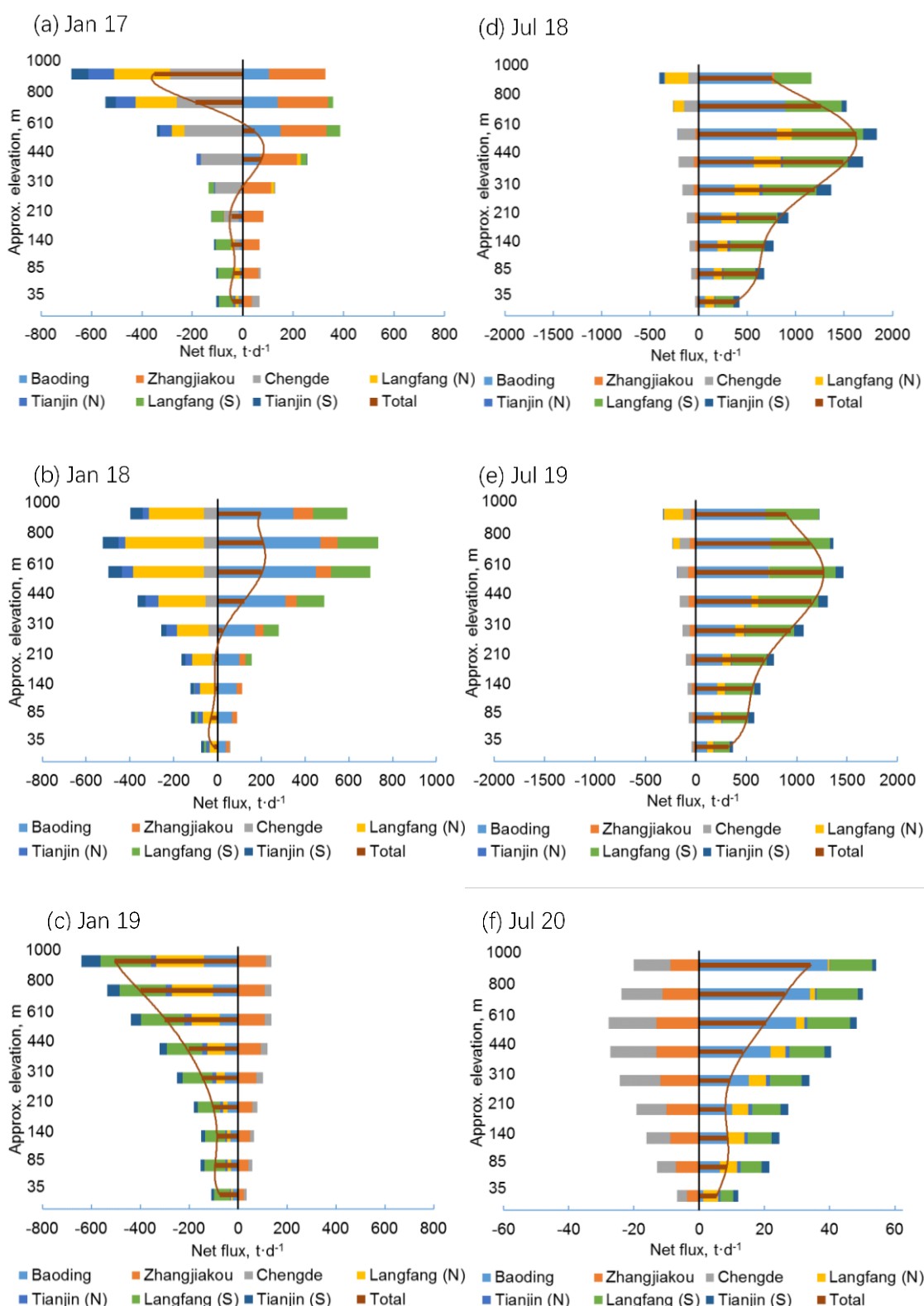


**Figure 9 PM$_{2.5}$ fluxes during heavy-pollution days in Beijing in January and July: (a) January 17th, (b) January 18th, (c) January 19th, (d) July 18th, (e) July 19th and (f) July 20th.**