# Peer review of "Assessment of inter-city transport of particulate matter in the Beijing-Tianjin-Hebei region"

_Atmospheric Chemistry and Physics, 2017_

## Referee Comment (RC1) · Anonymous Referee #2 · 25 Dec 2017

This paper applied the WRF-CMAQ model to simulate the air quality over the Beijing-Tianjin-Hebei area and calculate the trans-boundary fluxes to Beijing, Tianjin, and Shijiazhuang. This paper used a new method that assessed the pollutants transport by vertical surface flux calculation instead of usually used scenario analysis. I only have a few comments:

(1) Using this method, it is easily to understand the pollutants inflows from each direction or each surrounding city to the objective city. But the inflows from one city didn't necessarily mean those pollutants were from that city. It might be generated from the upflow cities. Did the authors consider about that? And is there any consideration about solving this? (2) Wind directions are very important to calculate the pollutants fluxes (Figure 3 and Equation 1). But in the model evaluation section (S2), the authors

only evaluate the wind speed, temperature, and humidity. I would suggest the authors to evaluate the wind direction in the simulation results.

---

## Short Comment (SC1) · 15 Jan 2018

The manuscript is meaningful for the prevention and control of regional pollution in north China. It is absolutely worth of publishing as the study itself is extremely interesting. However, some improvements are suggested.

In the manuscript, the authors found the southwest-northeast transport pathway. Actually, it is the most important pathway in North China Plain, especially during the heavy polluted episodes. Tang et al. (2015) and Zhu et al. (2016) found aerosols transported from the southwest between 500-1200 m (in the upper boundary layer) using ceilometer observations, which were the same with your simulations. However, the transport just emerged during the initial periods of the heavy pollution episodes. With the increase of the aerosols, the PBL decreases (below 500m) and the transport effects weaken during the heavy polluted periods. Could you please quantify the transport in different pollution degrees? In addition, some precursors were also transported in the initial periods. Afterwards, the precursors will react and form particles. Could you please quantify the contributions of the particles and the precursors' transport?

What's more, without the passage of large- or medium-scale meteorological system, the local mountain-plain winds emerges in North China Plain (Tang et al., 2016, Fig. 10). The alternation between the mountainous (northeast) winds that begin at 03:00 LT at night and the plain (southwest) winds that begin at 12:00 LT in the afternoon occurs. Therefore, air pollutants will transport to the northeast direction in the afternoon and then transport back during latter of half of the night. Could you please clarify the transport circulations combined with the influences of the mountain-plain winds?

Tang, G., Zhang, J., Zhu, X., Song, T., Münkel, C., Hu, B., Schäfer, K., Liu, Z., Zhang, J., Wang, L., Xin, J., Suppan, P., and Wang, Y.: Mixing layer height and its implications for air pollution over Beijing, China, Atmos. Chem. Phys., 16, 2459-2475, doi:10.5194/acp-16-2459-2016, 2016. Tang, G., Zhu, X., Hu, B., Xin, J., Wang, L., Münkel, C., Mao, G., and Wang, Y.: Impact of emission controls on air quality in Beijing during APEC 2014: lidar ceilometer observations, Atmos. Chem. Phys., 15, 12667-12680, doi:10.5194/acp-15-12667-2015, 2015. Zhu, X. Tang, G., Hu, B., Wang, L., Xin, J., Zhang, J., Liu, Z., Munkel, C., and Wang, Y.: Regional pollution and its formation mechanism over North China Plain: A case study with ceilometer observations and model simulations, J. Geophys. Res. Atmos., 121, 14574-14588, doi: 10.1002/2016JD025730, 2016.

---

## Referee Comment (RC2) · Anonymous Referee #1 · 21 Jan 2018

This paper analyzed the flux flow between cities in BTH area, northern plain in China, with a commonly used transport model WRF-CMAQ. It is an important issue for policy makers to understand the regional transport of air pollution, and would be helpful in decision of emission control strategy. The paper is clearly written and easy to follow. I suggest its publication when the following issues are further stressed or discussed.

1. Language. There are some grammar errors in the manuscript and the language should be polished.

2. Lines 56-57, Page 3. It is not quite persuasive, since the non-linear relationship is considered in the DDM and RSM methods.

3. Lines 128-129, Page 6. The authors stated that the biases of simulated meteoro-

logical field and PM2.5 concentrations fall in a reasonable range. For meteorological field, the statement could be supported by Table S1, with the evidence by Emery et al., 2001. For PM2.5 concentrations, however, we could not think it is "reasonable" as no further information is given. The bias could be quite large in some case, and some major components such as SOC were largely underestimated as indicated by the authors. Therefore, I suggest that the authors provide the evidence or criterion to justify the model performance., or describe the current simulation progress (model performance) in BTH region.

4. Section 3.2, Page 9. The authors described the difference in flux pattern between Jan and July. However, the reasons for the difference is not further discussed, and the seasonal mechanisms in pollution transport remained unclear. More information should be provided here.

5. Related with Q4, the paper described the pattern of pollution transport between cities, which is helpful for policy making. For scientific issue, however, the main factors influencing the transport were not sufficiently discussed. Could the author explain the roles of emissions and meteorological condition on the transport using the cases presented in the paper?

---

## Author Comment (AC1) · 5 Mar 2018

Please find the attached documents for the reply and revised manuscripts

Please also note the supplement to this comment:
https://www.atmos-chem-phys-discuss.net/acp-2017-933/acp-2017-933-AC1-supplement.zip

---

## Author Comment (AC2) · 5 Mar 2018

Please find the attached for the reply and revised manuscripts.

Please also note the supplement to this comment:
https://www.atmos-chem-phys-discuss.net/acp-2017-933/acp-2017-933-AC2-supplement.zip

---

## Author Comment (AC3) · 5 Mar 2018

Please check the attached for the response and the revised manuscripts.

Please also note the supplement to this comment:
https://www.atmos-chem-phys-discuss.net/acp-2017-933/acp-2017-933-AC3-supplement.zip
* * *

---

## Author Response (AR1)

RC1:

This paper applied the WRF-CMAQ model to simulate the air quality over the Beijing-Tianjin-Hebei area and calculate the trans-boundary fluxes to Beijing, Tianjin, and Shijiazhuang. This paper used a new method that assessed the pollutants transport by vertical surface flux calculation instead of usually used scenario analysis.

Author's reply: We feel encouraged to receive the reviewer's recognition for our work. We treasure the valuable comments raised by the referee and have followed them in revising the manuscript. Please check the below for the point-to-point responses.

(1) Using this method, it is easily to understand the pollutants inflows from each direction or each surrounding city to the objective city. But the inflows from one city didn't necessarily mean those pollutants were from that city. It might be generated from the upflow cities. Did the authors consider about that? And is there any consideration about solving this?

Authors' reply: We appreciate for the valuable comment. Indeed, the transport flux itself could not tell whether the pollutants are from the neighbor city or from other upstream cities, and this problem is one of the major disadvantages of the flux method. However, the goal of using the flux method is mainly focused on the transport from different directions, rather than the contribution of different cities. By putting together the transport characteristics of different cities as is done in our study (see Fig. R1), we have obtained a general transport feature in the BTH region, which can facilitate a qualitative understanding of where the fluxes are mainly from.

As shown in Fig. R1, in January, the $PM_{2.5}$ that flows into Beijing from Baoding on the southeast mainly happens at a higher level, so the $PM_{2.5}$ may origin from a larger area upstream. Then we can track backward along the dark arrow, and the inflow may come from Baoding, Shijiazhuang or even Xingtai. In July, the transport directions between Baoding and Shijiazhuang are different at the lower and the upper level. We can infer that the inflowing $PM_{2.5}$ into Shijazhuang mainly come from Baoding rather than farther regions to the northeast.

We admit that the flux approach cannot quantitatively evaluate the contribution from each city, and we hope that future studies can combine the flux method with other methods such as tagging models to quantitatively assess the contribution of different cities or regions on the transport pathways identified in the current study.

We have included the preceding discussions on the limitations in the revised manuscript. (Page 15, Line 401-407)

[Figure]

Figure R1 The transport fluxes through each boundary segment of the three target cities in January (a) and July (b). The size of the arrows represents the amount of the fluxes, while white and black arrows denote fluxes at the lower (layer 1-5 in the model, from the ground to about 310 m) and upper (layer 6-9 in the model, from about 310 m to about 1000 m) layers, respectively.

(2) Wind directions are very important to calculate the pollutants fluxes (Figure 3 and Equation 1). But in the model evaluation section (S2), the authors only evaluate the wind speed, temperature, and humidity. I would suggest the authors to evaluate the wind direction in the simulation results.

Authors' reply: We thank the referee for the good suggestion. We evaluated the wind direction using the same method as the other meteorology parameters. The results are shown in Table R1. The bias of wind direction at 10m (WD10) falls within the benchmark, but the gross error exceeds the benchmark for both January and July. The larger gross error is partly caused by the lower precision of the observation data. The WD10 observations only have 16 different values, while the simulation could have any value between 0 and 360. For example, if the real WD10 is 125 degree, the value will be reported as 140 degree. Even if the simulation is exactly 125 degree, an additional gross error of 15 degree will be introduced. In addition, compared to other similar simulation studies in China (e.g., Hu et al., 2016; Zhao et al., 2013), the gross error of WD10 falls in a similar range. Therefore, we still believe that the simulated meteorology field shows a reasonable agreement with the observation.
We have added the data and the discussion above in our SI (Page 3, Line 37 − 50).

Table R1 Comparison of simulated and observed wind direction

| Parameter | Index | Unit | Benchmark[a] | Jan-2012 | Jul-2012 |
|---|---|---|---|---|---|
| | Observation Mean | deg | - | 203.9 | 175.2 |
| | Simulation Mean | deg | - | 222.4 | 174.8 |
| Wind direction (WD10) | Bias | deg | ≤±10 | -2.64 | -1.47 |
| | Gross error | deg | ≤30 | 43.23 | 43.7 |

a. The benchmarks used in this study are suggested by Emery (2011).

RC2:
This paper analyzed the flux flow between cities in BTH area, northern plain in China, with a commonly used transport model WRF-CMAQ. It is an important issue for policy makers to understand the regional transport of air pollution, and would be helpful in decision of emission control strategy. The paper is clearly written and easy to follow. I suggest its publication when the following issues are further stressed or discussed.

Author's reply: We appreciate the reviewer's valuable comments which help us improve the quality of the manuscript. We have carefully revised the manuscript according to the reviewers' comments. Below is our point-to-point responses to the issues raised by the reviewer.

(1) Language. There are some grammar errors in the manuscript and the language should be polished.

Authors' reply: We sincerely apologize for the deficiency in language. We have gone through the text carefully and corrected the grammar errors.

(2) Lines 56-57, Page 3. It is not quite persuasive, since the non-linear relationship is considered in the DDM and RSM methods.

Authors' reply: We thank the referee for pointing out the problem. The expression in the manuscript was not quite accurate and might cause misunderstanding. We believe that all methods based on chemical transport models, including the DDM and RSM methods, are able to consider the non-linear relationship between emissions and concentrations. However, the sensitivity of $PM_{2.5}$ concentrations to emission perturbation, which DDM and RSM aim to quantify, is different from the inter-city transport of $PM_{2.5}$. Assuming that the emission reduction in the source region leads to a 30% reduction in $PM_{2.5}$ concentration in the target region, the transboundary transport of $PM_{2.5}$ may not be 30% because of the nonlinearity in the emission-concentration relationships.
Zhao et al (2017) found that during winter time, the $PM_{2.5}$ could response negatively to NOx reduction. Therefore, if we use sensitivity approach such as brute-force method,

DDM and RSM to assess the regional transport of $PM_{2.5}$, the result may deviate (probably underestimate) from the real case.

We changed our expression in the manuscript for a better understanding of the insufficiency of the DDM and RSM methods for our study (Page 3, Line 55-57).

(3) Lines 128-129, Page 6. The authors stated that the biases of simulated meteorological field and $PM_{2.5}$ concentrations fall in a reasonable range. For meteorological field, the statement could be supported by Table S1, with the evidence by Emery et al., 2001. For $PM_{2.5}$ concentrations, however, we could not think it is "reasonable" as no further information is given. The bias could be quite large in some case, and some major components such as SOC were largely underestimated as indicated by the authors. Therefore, I suggest that the authors provide the evidence or criterion to justify the model performance., or describe the current simulation progress (model performance) in BTH region.

Authors' reply: We thank the referee for the valuable suggestion. We calculated additional indices including Mean Fractional Bias (MFB) and Mean Fractional Error (MFE), and compared them with the benchmark values suggested by Boylan and Russell (2006). The definition of the two indices can also be found in the research of Boylan and Russell (2006). The simulation results of the $PM_{2.5}$ concentration are well within the model performance criteria. We have added the new statistical results as well as the benchmark values in Table 2 in the manuscript.

Due to the limitation of monitoring sites, the benchmark is not applicable for the evaluation of $PM_{2.5}$ components. Therefore, we compare the model performance with other studies in the BTH region. The results are summarized in Table R2. All of the studies underestimate the sulfate concentrations. The underestimation ranges between 9% and 79%, and most of them are larger than 30%. The nitrate simulation results vary in different studies, but the majority of the studies tend to overestimate its concentration. The concentration of EC is usually much lower than the other four components, which may contribute to the large discrepancy in the simulation results in different studies. For OC, although some studies overestimate the concentration, more studies exhibit a lower concentration than observation. Generally speaking, the biases of the $PM_{2.5}$ components in the current study have similar magnitude to other recent studies in the BTH region.

We have included the discussion in the revised SI (Page 7, Line 69-80)

Table R2 Summary of the PM$_{2.5}$ component simulation results for the BTH region in recent studies

| Time | Site | SO$_4^{2-}$ NMB (%) | NO$_3^-$ NMB (%) | NH$_4^+$ NMB (%) | EC NMB (%) | OC NMB (%) | Reference |
|---|---|---|---|---|---|---|---|
| 2005 annual | Tsinghua, Beijing | -14 | 13 | 10 | -24 | -36 | Wang et al., 2011 |
| 2005 annual | Miyun, Beijing | -36 | 62 | 9 | -17 | -52 | Wang et al., 2011 |
| 14 Jan – 8 Feb, 2010 | Beijing | -72 | -32 | -5 | 124 | 26 | Liu et al., 2016 |
| 14 Jan – 8 Feb, 2010 | Shangdianzi, Beijing | -78 | -24 | -13 | 36 | -7 | Liu et al., 2016 |
| Jan, 2010 | Peking university, Beijing | -39 | 85 | 33 | 101 | -2 | Liu et al., 2016 |
| 14 Jan – 8 Feb, 2010 | Shijiazhuang, Hebei | -79 | -35 | -7 | 81 | 38 | Liu et al., 2016 |
| 14 Jan – 8 Feb, 2010 | Chengde, Hebei | -78 | 48 | -10 | -39 | -50 | Liu et al., 2016 |
| 14 Jan – 8 Feb, 2010 | Tianjin | -72 | 0 | 9 | 149 | 85 | Liu et al., 2016 |
| 11-15 Jan, 2013 | Beijing | ~ -73 | ~ -43 | - | - | - | Wang et al., 2014 |
| Jan 2013 | Handan, Hebei | -9 | 33 | -11 | 50 | 37 | Wang et al., 2015 |
| Jul 2013 | Handan, Hebei | -32 | -3 | 8 | 96 | 30 | Wang et al., 2015 |
| Oct – Nov, 2014 | 7 sites in the BTH region | -48 | 16 | -25 | 87 | -37 | Zhao et al., 2017 |
| 22 Jul – 23 Aug, 2012 | Xiong County, Hebei | -52 | 95 | 2 | 120 | -25 | This study |
| 22 Jul – 23 Aug, 2012 | Ling County, Shandong | -57 | 79 | -14 | 117 | -1 | This study |

(4) Section 3.2, Page 9. The authors described the difference in flux pattern between Jan and July. However, the reasons for the difference is not further discussed, and the seasonal mechanisms in pollution transport remained unclear. More information should be provided here.

Authors' reply: We appreciate for the valuable comment. What determine the seasonal transport flux are mainly two factors, the wind speed and the PM$_{2.5}$ concentration in the upstream areas. The PM$_{2.5}$ concentration is related to both the meteorology condition and the emission, with the upstream emissions being the most important factor. If we understand the roles of emissions and winds in the transport, we can answer the question of the seasonal mechanisms. Therefore, we combined the response of this comment with the fifth one below.

(5) Related with Q4, the paper described the pattern of pollution transport between cities, which is helpful for policy making. For scientific issue, however, the main factors influencing the transport were not sufficiently discussed. Could the author explain the roles of emissions and meteorological condition on the transport using the cases presented in the paper?

Authors' reply: We combine the response of this comment with the fourth one. To better understand how the wind and concentration affect the transport fluxes, we have made several wind rose plots for different cities, different seasons and different heights. Besides the traditional wind rose plot that displays wind direction with wind speed frequencies, we also made plots that display the wind direction with PM$_{2.5}$

concentration frequencies. We chose the ground layer and the 7th layer to represent the lower layer and the upper layer respectively. The plots for Beijing are shown in Fig. R2 as an example. The plots for the other two cities are displayed in the SI (Fig. S3 and Fig. S4).

In January, the dominant wind directions near the ground ranges from northwest to northeast. The NNE wind has the highest frequency, while the NW wind has the highest wind speed (Fig. R2(a)). The dominant northern winds reflect the winter monsoon. Although the concentration coming with the northern winds are relatively low because of the low emission rate on that direction (Fig. R2(b)), the high frequency and wind speed also cause an overall strong transport from the northwest to the southeast. Wind directions and the corresponding concentrations are quite different at the upper layers (Fig. R2(c), (d)). The prevalent northern wind remains (though the dominant directions shift slightly from NNE to NW), and the frequency of southwestern winds is much higher than that at lower layers. Moreover, the $PM_{2.5}$ concentrations that come with southwestern winds are much higher than the other directions. The strong emission in southern Hebei (which lies on the southwest direction of Beijing), especially the elevated sources may be responsible for the high concentration from the southwest. Therefore, in January, the dominant northwestern winds account for the Northwest-Southeast pathway at both lower layers and upper layers, while the large emissions on the southwest direction mainly caused the Southwest-Northeast pathway at upper layers.

In July, the dominant wind directions at the lower layer are the southeastern directions, reflecting the summer monsoon (Fig R2(e)), and coincidentally the highest concentrations also come along with the southeastern winds (Fig R2(f)). Emissions from Tianjin, Langfang, and Tangshan may influence Beijing by the southeastern winds. The emission and the wind direction both contribute to the Southeast-Northwest pathway at the lower layers. The high frequency wind directions shift clockwise to the southern directions at the upper layers in July, as is shown in Fig. R2(g), and the southwest wind and the southeast wind are both important. Moreover, the directions with high concentrations also shift to both the southwest and the southeast directions (Fig. R2(h)). Therefore, in July, the dominant southeastern winds and the emissions on the southeast directions caused the Southeast-Northwest pathway at both the upper and the lower layers. The Southwest-Northeast pathway is a combination result from the southern winds and the emissions, which is different from that in January.

Similar analysis can be made to the plots for the other two cities. Due to the length limitation, we put the plots into the SI. The plots in Fig. R2 and the discussions above has been included in our revised manuscript (Page 10-11, Line 262-287).

[Figure]

Figure R2 The wind rose plots showing the frequency of wind speed (a, c, e, g) and PM₂.₅

SC1:
The manuscript is meaningful for the prevention and control of regional pollution in north China. It is absolutely worth of publishing as the study itself is extremely interesting. However, some improvements are suggested.

Authors' reply: It is our honor to receive the valuable comments from Dr. Tang. We have revised the manuscript carefully according to these comments. Please see below for our point-to-point responses.

In the manuscript, the authors found the southwest-northeast transport pathway. Actually, it is the most important pathway in North China Plain, especially during the heavy polluted episodes. Tang et al. (2015) and Zhu et al. (2016) found aerosols transported from the southwest between 500-1200 m (in the upper boundary layer) using ceilometer observations, which were the same with your simulations. However, the transport just emerged during the initial periods of the heavy pollution episodes. With the in crease of the aerosols, the PBL decreases (below 500m) and the transport effects weaken during the heavy polluted periods. Could you please quantify the transport in different pollution degrees?

Authors' reply: The reviewer raised a very useful question. The transport fluxes vary with different meteorology conditions in different days. Following this suggestion, we calculated the flux for individual days in January and July, and sorted the data into groups based on different pollution levels (see Fig. R3). Taking Beijing as an example, in January, the simulated concentration ranges from 11 $\mu g/m^3$ to 271 $\mu g/m^3$, while in July, the range is from 6 $\mu g/m^3$ to 94 $\mu g/m^3$. We set 6 groups for January and 5 groups for July. The separating points are chosen to be near the 30, 55, 75, 85 and 95 percentiles in January, and the 30, 60, 80, 90 percentiles in July. The groups are denser at higher concentrations to better reveal the details before and after heavy pollution periods.
In January, the transport becomes stronger when the concentration is higher, but the transport flux decreases in turn when the concentration is the highest. The inflow from Baoding and outflow to Chengde, which are the indicator of the Southwest-Northeast pathway, also experience a gradual rise followed by a sudden decline. In July, the situation is similar, though the decrease is less significant. Such result is consistent with Tang et al. (2015) and Zhu et al. (2016) that the Southwest-Northeast transport pathway is more significant during the rising phase of a heavy pollution period, but fades when the pollution reaches the peak.
Inspired by these results, we also conducted a day-to-day analysis on the two heavy pollution episodes described in Section 3.3, which occur in January and July, respectively (Fig. R4). We find the "flowing in and accumulating" phenomenon for both episodes. For the episode in January, the inflow (especially from southwest) is strong in January 18th, while the inflow declines rapidly in January 19th, the day with the highest concentration. The phenomenon is more significant during the episode in July. In July 18th and 19th, the inflow flux is very strong, while in July 20th which has the highest concentration, the flux decreases for more than one order of magnitude. This finding emphasizes the importance of early temporary control before heavy pollution occurs. We have revised our manuscript to include the above results and discussions. (Page 11-12, Line 294-306)

[Figure]

Figure R3 PM$_{2.5}$ average flux between Beijing and its neighboring cities in different pollution degrees in (a) January and (b) July.

[Figure]

Figure R4 PM$_{2.5}$ fluxes during heavy-pollution days in Beijing in January and July: (a) January 17th, (b) January 18th, (c) January 19th, (d) July 18th, (e) July 19th and (f) July 20th.

In addition, some precursors were also transported in the initial periods. Afterwards, the precursors will react and form particles. Could you please quantify the contributions of the particles and the precursors' transport?

Authors' reply: The transport of precursors that may transform into particles is indeed an important factor. However, only if the precursors are tracked in all the physical and chemical reactions can we quantify the contribution of the precursor's transport to the $PM_{2.5}$. The flux approach is not able to account for this issue, which is one of the main shortages. Nevertheless, the flux approach can capture the transport features of all primary and most of the secondary $PM_{2.5}$. We have some discussion on this shortage in our manuscript, and we hope that future study can combine the tracer model with the flux approach to overcome this shortage.

What's more, without the passage of large- or medium-scale meteorological system, the local mountain-plain winds emerges in North China Plain (Tang et al., 2016, Fig. 10). The alternation between the mountainous (northeast) winds that begin at 03:00 LT at night and the plain (southwest) winds that begin at 12:00 LT in the afternoon occurs. Therefore, air pollutants will transport to the northeast direction in the afternoon and then transport back during latter of half of the night. Could you please clarify the transport circulations combined with the influences of the mountain-plain winds?

Authors' reply: We thank the reviewer very much for this useful comment. In our original study, we calculated daily $PM_{2.5}$ fluxes, so that the mountain-plain winds (which is a diurnal variation feature) is not taken the into consideration. Following the reviewer's comment, we tried to probe into the diurnal wind and flux pattern in Beijing. The simulated average diurnal wind patterns at 100 m height in January and July in Beijing are shown in Fig. R5(b). We also put the observation results from Tang et al., (2016) in Fig R5(a) as a reference. We find that the simulated wind pattern is consistent with the observation. In January, the mountain-plain winds are presented as the change in wind speed, but the wind direction does not change significantly during the whole day. In July, there is a significant wind direction shift, similar to the description of the reviewer. The mountainous wind (northeast) begins at 2:00 LT, and is taken over by the plain wind (southeast) at about 10:00 LT, and the mountainous wind is much weaker than the plain wind. A circulation of mountain-plain wind may have influence on the transport of $PM_{2.5}$ in July.

Considering that the mountain-plain wind circulation mainly happens at the foot of the mountains, we calculated the fluxes through the boundaries between Beijing and its three neighboring cities on the south/southeast (Baoding, Langfang and Tianjin) during mountainous wind hours and the plain wind hours in July separately (Fig. R6). During the plain wind hours, all the boundaries on the southwest and southeast of Beijing have positive net fluxes, which is due to the relatively strong southerly plain winds. During the mountainous wind hours, however, there is no significant direction change of the fluxes except for the boundary of Baoding and Southern Langfang at levels below 200 m. The sign of fluxes mostly remains unchanged because the mountain-plain wind circulation is weaker at higher levels, and the wind speed of the mountainous wind is even weaker at the southernly boundaries which has limited effect to alter the sign of the flux. Nevertheless, the magnitude of fluxes is significantly smaller than the plain wind hours, which is partly attributed to the mountain-plain wind circulation. Therefore, the summertime mountain-plain wind circulation in Beijing does not significantly alter the sign of inter-city $PM_{2.5}$ fluxes but does have considerable impact on their magnitude. We have included the discussion on the mountain-plain wind in our revised manuscripts (Page 14, Line 365-369 in the main text, and Page 12-13, Line 102-131 in SI).

[Figure]

Figure R5 The observed and simulated monthly average diurnal variation of winds in Beijing in July. (a) The observation results from Tang et al. (2016). (b) The simulation results in this study.

[Figure]

Figure R6 The transport fluxes in July between Beijing and its neighboring cities during (a) plain wind hours (11:00 – 1:00 (+1 day) LT) and (b) mountainous wind hours (2:00 – 10:00 LT)

[revised manuscript text omitted]